# T Cell Responses Correlate with Self-Reported Disease Severity and Neutralizing Antibody Responses Predict Protection against SARS-CoV-2 Breakthrough Infection

**DOI:** 10.3390/v15030709

**Published:** 2023-03-09

**Authors:** Zhen Zhao, Attila Kumanovics, Tanzy Love, Stacy E. F. Melanson, Qing H. Meng, Alan H. B. Wu, Joesph Wiencek, Fred S. Apple, Caitlin R. Ondracek, David D. Koch, Robert H. Christenson, Yan Victoria Zhang

**Affiliations:** 1Department of Laboratory Medicine and Pathology, Weill Cornell Medicine, New York, NY 10065, USA; 2Department of Laboratory Medicine and Pathology, Mayo Clinic, Rochester, MN 55905, USA; 3Department of Biostatistics and Computational Biology, University of Rochester, Rochester, NY 14642, USA; 4Department of Pathology, Brigham and Women’s Hospital, Boston, MA 02115, USA; 5Harvard Medical School, Boston, MA 02115, USA; 6Department of Laboratory Medicine, The University of Texas MD Anderson Cancer Center, Houston, TX 77030, USA; 7Department of Laboratory Medicine, University of California, San Francisco, CA 94143, USA; 8Department of Pathology, Microbiology and Immunology, Vanderbilt School of Medicine, Nashville, TN 37240, USA; 9Department of Laboratory Medicine and Pathology, Hennepin Healthcare/Hennepin County Medical Center, Minneapolis, MN 55415, USA; 10Hennepin Healthcare Research Institute, Minneapolis, MN 55404, USA; 11American Association for Clinical Chemistry, Washington, DC 22203, USA; 12Department of Pathology and Laboratory Medicine, Emory University, Atlanta, GA 30303, USA; 13Department of Pathology, University of Maryland School of Medicine, 685 W. Baltimore Street, Baltimore, MD 21201, USA; 14Department of Pathology and Laboratory Medicine, University of Rochester Medical Center, 601 Elmwood Avenue, Box 626, Rochester, NY 14642, USA

**Keywords:** COVID-19, SARS-CoV-2, vaccine, T cell response, neutralizing antibody, breakthrough infection

## Abstract

Objectives: The objective of this prospective study was to investigate the role of adaptive immunity in response to SARS-CoV-2 vaccines. Design and Methods: A cohort of 677 vaccinated individuals participated in a comprehensive survey of their vaccination status and associated side effects, and donated blood to evaluate their adaptive immune responses by neutralizing antibody (NAb) and T cell responses. The cohort then completed a follow-up survey to investigate the occurrence of breakthrough infections. Results: NAb levels were the highest in participants vaccinated with Moderna, followed by Pfizer and Johnson & Johnson. NAb levels decreased with time after vaccination with Pfizer and Johnson & Johnson. T cell responses showed no significant difference among the different vaccines and remained stable up to 10 months after the study period for all vaccine types. In multivariate analyses, NAb responses (<95 U/mL) predicted breakthrough infection, whereas previous infection, the type of vaccine, and T cell responses did not. T cell responses to viral epitopes (<0.120 IU/mL) showed a significant association with the self-reported severity of COVID-19 disease. Conclusion: This study provides evidence that NAb responses to SARS-CoV-2 vaccination correlate with protection against infection, whereas the T cell memory responses may contribute to protection against severe disease but not against infection.

## 1. Introduction

The COVID-19 pandemic has affected every facet of life across the globe for more than two years. As vaccines become more readily available and vaccination rates improve, both public and scientific communities are eager to learn more about the effectiveness of the various vaccines as objectively measured through humoral and cellular immune responses.

Long-term immunological memory to SARS-CoV-2 vaccines is crucial for the development of population-level immunity. Previous reports on the rapid waning of SARS-CoV-2 antibody levels and the loss of neutralizing capacity against the Delta, Omicron, and other Variants of Concern (VoC) [1,2], as well as the occurrence of minimally to moderately affected vaccine efficacy, have led to questions regarding the efficacy of humoral immunity post natural infection and vaccination [3]. In contrast, functional T cell responses in both frequency and intensity remained robust after one year post natural infection, and at least six months post vaccination [4,5]. SARS-CoV-specific T cells were still detectable 17 years after infection [6]. T cell-mediated immunity is thought to play a critical role in the immune response to SARS-CoV-2 infection, in preventing severe disease after natural infection. However, it is difficult to decipher the level of humoral and cellular immune responses required to protect against infection or severe disease [7]. Despite the reports of largely preserved vaccine-induced T cell response against VoCs [1,2] and minimally to moderately affected vaccine efficacy against VoCs in protecting individuals from severe disease, hospitalization, and death [8,9,10], the impact of T cell responses against various SARS-CoV-2 outcomes, such as breakthrough infection and severe disease, in vaccinated populations is currently unknown [7].

While many studies have investigated the durability of immune responses to SARS-CoV-2 vaccines, to our knowledge, studies that include large, diverse populations with various vaccines administered for evaluations of both humoral and cellular immune correlates of protection have not been conducted. The objective of this prospective study was to investigate whether neutralizing antibody (NAb) and T cell responses induced by different, widely used SARS-CoV-2 vaccines are durable and protective in a large, diverse cohort, as well as to illustrate the utility of NAb and T cell responses in protecting against SARS-CoV-2 infection and severe disease.

## 2. Methods

### 2.1. Study Design

This study was organized by the American Association for Clinical Chemistry (AACC) and included an online survey available between 9 September and 20 October 2021, as well as an onsite blood sample collection from 27 to 30 September 2021, during the AACC annual scientific meeting. AACC members were informed via email and/or social media about the study enrollment. The survey was designed to gather information from laboratory professionals about COVID-19 vaccination and its side effects, and these findings were recently published [11]. A follow-up survey (Appendix A) was sent to participants four months after the blood collection that took place from 9 to 23 February 2022, in order to gather information on breakthrough infection [12]. The study was approved by the University of Maryland Institutional Review Board and informed consent was obtained for participants.

### 2.2. Inclusion and Exclusion Criteria

AACC members and conference attendees were invited to participate [11]. The only exclusion criteria were age (<18 years) and pregnancy. Due to the heterogeneity and relatively small sample size, participants who received other or unknown vaccine types and those that had both a previous infection and a booster were not included in the analysis. There were 677 subjects included in the analysis dataset (Figure 1).

### 2.3. Specimen Collection and Storage

Venous blood was collected into a serum separator tube (BD 368774), fully clotted, and then centrifuged (1500× *g* for 15 min at room temperature) within two hours of collection. Serum samples were aliquoted, stored at 4 °C at the sample collection site, and then transported to the Centers for Disease Control and Prevention (CDC) within four hours for aliquoting on TECAN (Mannedorf, Switzerland). The samples were stored in 350 µL aliquots at −80 °C until analysis (Eppendorf CryoStorage Vials, 0.5 mL, Mfr. No. 0030079400). Whole blood was collected into a lithium–heparin tube and stored at 4 °C for up to eight hours prior to overnight shipment to the testing clinical laboratory (Quest Diagnostics, Secaucus, NJ, USA) for the T cell immunity analysis, according to the manufacturer’s guidelines [13,14]. The testing was performed at time points consistent with the samples’ collection time points across the three-day sample collection period.

### 2.4. Laboratory Analysis

#### 2.4.1. Anti-SARS-CoV-2 Nucleocapsid (Anti-N) Antibody Assay

The Platelia SARS-CoV-2 Total Ab Enzyme-Linked Immunosorbent Assay (ELISA) assay (Bio-Rad Laboratories, Hercules, CA, USA) is a one-step antigen (Ag)-capture format ELISA, used for the qualitative detection of total anti-N antibodies (IgM/IgG/IgA) in human serum or plasma specimens. This assay received the FDA’s Emergency Use Authorization (EUA) [15]. The assay was performed according to the manufacturer’s instructions at a clinical laboratory at the University of Maryland.

#### 2.4.2. SARS-CoV-2 NAb Assay

Semi-quantitative detection of SARS-CoV-2 total NAb was performed using the GenScript cPass SARS-CoV-2 Neutralization Antibody Detection Kit (GenScript, Piscataway, NJ, USA), which is a functional ELISA kit that received the FDA’s EUA [16]. The assay was performed according to the manufacturer’s instructions at Quest Diagnostics (Chantilly, VA, USA). The cutoff value for the cPass SARS-CoV-2 NAb Detection Kit is 47 Units/mL (30% signal inhibition). The test was calibrated for the semi-quantitative detection of anti-SARS-CoV-2 NAb by using the SARS-CoV-2 NAb Calibrator (GenScript, Piscataway, NJ, USA).

#### 2.4.3. QuantiFERON SARS-CoV-2 Assay

The QuantiFERON 2 plate ELISA kit (Qiagen, Hilden, Germany) was used on a Dynex Agility (Dynex Technologies Inc., Chantilly, VA, USA) at Quest Diagnostics (Chantilly, VA, USA). Since this assay was allowed for research use only at the time of analysis, the laboratory performed an internal validation (Appendix A). The Limit of detection (LOD) and Limit of quantitation were established as 0.038 IU/mL and 0.061 IU/mL interferon-gamma (INF-γ), respectively. The LOD was applied as the cutoff value for a positive response.

The sample tubes were incubated at 36–38 °C for 16–24 h and centrifuged for 15 min at 2000–3000× *g*. The steps included the transfer of samples to the QuantiFERON SARS-CoV-2 Starter Set Blood Collection Tubes (QFN SARS-CoV-2 BCTs, consisting of SARS-CoV-2 Ag1 and SARS-CoV-2 Ag2 tubes), the QuantiFERON SARS-CoV-2 Extended Set BCTs (QFN SARS-CoV-2 Extended Set BCT, consisting of SARS-CoV-2 Ag3 tubes), and the QuantiFERON Control Set BCTs (Qiagen, Hilden, Germany), according to the manufacturer’s guidelines [13] and as described previously [17]. Following ELISA, background INF-γ levels were subtracted to obtain quantitative results (INF-γ in IU/mL) for analysis. Further details are supplied in the Appendix A.

### 2.5. Data Analysis

Basic demographic information was compared using descriptive statistics. Previous positive COVID-19 cases were defined based on self-reported positive PCR/Ag tests and/or a positive Platelia SARS-CoV-2 Total Ab ELISA assay (Anti-N antibody assay). Comparisons were made using the Kruskal–Wallis rank sum ANOVA (for continuous data) or Chi-squared tests (for categorical data). With 677 subjects, there was 80% power to compare differences of at least a 0.15 effect size (small–moderate) between vaccine types, boosters, and previous infections.

Linear models were fit, expressing the change in log biomarker values over months since the last dose. Estimates of the change in log biomarker values from the reference (two to four months since last dose) and their significance are reported. Univariate models include only time, since the vaccine dose and multivariate models also include sex, age, and the number of self-reported preexisting conditions as covariates, along with the other multivariate models below. Excluding those boosted or with previous infections, there were 533 remaining subjects, which provides 80% power to detect changes in biomarker values over time that explain at least 3.5% of the variability (small effect size).

Breakthrough infection and its severity was self-reported in the follow-up survey. Comparisons of biomarker levels were made between subjects with different breakthrough infection severities using a one-way ANOVA. The mean biomarker levels for each severity group are reported with univariate and multivariate *p*-value testing for differences; multivariate models include age, sex, time from vaccination to blood being drawn, previous infection status before blood being drawn, and time after blood being drawn to breakthrough infection. The threshold for statistical significance was α = 0.05. Logistic regression was fit to predict the appearance of breakthrough infections and, separately, the severity of symptoms from vaccine type or biomarker assays. The prediction thresholds for the biomarker assays were either based on the LOD of the assay or were data-driven. For the data-driven approach, the thresholds were chosen by fitting a receiver operating characteristic (ROC) curve to the assay, to determine the optimal discriminator between those who had breakthrough infections in the follow-up period and between different symptom severity groups. We report the adjusted odds ratios between the two groups, in terms of the chance of each outcome and its significance, for both univariate and multivariate models; multivariate models include age, sex, time from vaccine to blood being drawn, previous infection status before blood being drawn, and time after blood being drawn to breakthrough infection. The 449 subjects who provided survey responses regarding breakthrough infections yielded 80% power to detect changes of at least 1.8 times the odds ratio (small effect size). To compare symptom severity, there were 87 positive cases, which provides 80% power to detect differences in severity of symptoms of at least six times the odds (large effect size).

## 3. Results

### 3.1. Participant Demographics and Characterization

Of the 698 people who participated in the on-site study, 677 were fully vaccinated and had completed the survey questionnaire at the time of sample collection. Among the 677 individuals, 564 indicated no previous infection (Pfizer, *n* = 314; Moderna, *n* = 181; Johnson & Johnson (J&J), *n* = 38; and booster, *n* = 31) and 113 reported a previous infection (Table 1). The 31 subjects that had received a booster at the time of sample collection collectively received 28 Pfizer–BioNTech (25 homologous and 3 heterologous), 2 Moderna (homologous), and 1 J&J (heterologous) booster doses. The previous infection group included 46 subjects that were positive for anti-N antibodies, and 52 subjects that were negative for anti-N antibodies; anti-N testing was not performed on the remaining 15 participants (Appendix A). The group with self-reported previous infection or positive anti-N response was not separated by vaccine type. There was no observable significant difference based on sex, age, race, or ethnicity among the five different groups (Table 1).

### 3.2. Follow-Up Survey

Sixty-six percent (449/677) of the fully vaccinated individuals (two-doses of a primary-series COVID-19 vaccine, or one dose of a single-dose, primary-series COVID-19 vaccine approved or authorized for use in the United States) participated in the follow-up February 2022 survey, of which, 66.1% (373/564) of participants in the infection-naïve group and 67.3% (76/113) of participants with previous infections completed the follow-up survey (Figure 1). In the follow-up survey, participants were asked if they tested positive after the blood draw; 18.0% (67/373) in the naïve group and 26.3% (20/76) in the previously infected group reported having breakthrough infections (positive Ag or PCR test). There was no significant difference in the rate of breakthrough infections between those who were infection-naïve vaccinated or previously infected and those who were vaccinated at the blood draw (*p* = 0.129). Among the breakthrough infections, 28.4%, 58.2%, and 13.4% (19, 39, and 9 out of 67) in the infection-naïve vaccinated group and 35.0%, 60.0% and 5.0% (7, 12, and 1 out of 20) in the previously infected and vaccinated group reported having infections with symptoms that caused no, mild/moderate, and severe limitations, respectively (Figure 1). The severity of the symptoms was based on definitions from the CDC (Vaccine Adverse Event Reporting System (VAERS), VAERS | Vaccine Safety | CDC) [18]. There was no significant difference in the severity of breakthrough infections between those who were infection-naïve and vaccinated or those who were previously infected and vaccinated at the blood draw (*p* = 0.552).

### 3.3. Post-Vaccination NAb and T Cell Responses at the Time of Sample Collection

NAb and T cell response were evaluated in the infection-naïve participants. Due to the small number of participants who had received a vaccine booster at that time (Table 1), they were not included in the analyses. In participants who were infection-naïve at the time of blood draw, 531 (99.6%) and 527 (98.7%) samples were measured for NAb and T cell responses, respectively. There was a poor correlation (r = 0.19–0.22) between T cell and NAb responses. Two to four months since the last dose of vaccination (PLDV) was used as a reference time window, due to the small number of individuals in the zero to two month window. At two to four months PLDV, levels of the measured assays were not significantly different in between individuals who received Pfizer and Moderna (*p* = 0.557, 0.342, 0.255, 0.199, 0.159 for the five different measures). However, those who received the J&J vaccine had significantly lower NAb levels (*p* < 0.001) (Appendix A). Compared to two to four months PLDV, log NAb levels were significantly decreased over time in participants after receiving Pfizer, and were moderately decreased after receiving J&J, but not after Moderna (Figure 2 and Appendix A). In contrast, the T cell responses to Ag1, Ag2, Ag3, and the sum of Ag 1–3 were not significantly different up to 10 months PLDV between Pfizer, Moderna, and J&J vaccines, except for sum of Ag 1–3 and Ag3, which were significantly higher at zero to two months PLDV compared to two to four months PLDV for Pfizer and/or J&J (*p* < 0.05). There was no significant difference for either the sum of Ag 1–3 or Ag3 after two to four months PLDV during the study period (Figure 2 and Appendix A). Similar results were also observed in multivariate models adjusted for sex, age, and the number of self-reported pre-existing conditions (Appendix A). In the models used to assess NAb responses, age is associated with a significant decrease in response (*p* < 0.001), while age is not a significant predictor of the T cell responses to Ag 1–3 (*p* = 0.255) (Appendix A).

### 3.4. NAb and T Cell Responses and Breakthrough Infection

Among the participants (*n* = 449) who answered the follow up survey questions, 19.4% (87/449) reported breakthrough infections with positive Ag or PCR tests. All of these individuals were vaccinated, but most (430/449) had only completed the primary vaccination series without booster shots at the time of blood draw.

T cell and Nab responses were evaluated between participants with or without breakthrough infection via univariate and multivariate analysis. No significant differences were detected between the two groups for these two assays (Table 2). Furthermore, no significant differences in these assays were observed comparing participants with breakthrough infections having different severity of symptoms (no, mild/moderate, severe) (Table 2, Appendix A).

Logistic regression was fit to predict the breakthrough infections and severity of symptoms. In the univariate model, those with NAb < 95 U/mL had 78% (*p* = 0.020) higher odds of having breakthrough infections at the time of the follow-up survey. Participants with T cell responses to Ag2 > 0.120 IU/mL had 13% (*p* = 0.045) lower odds of having severe limitations if they contracted a COVID-19 breakthrough infection at the time of the follow-up survey (Table 3). There was no significant difference in the prediction of breakthrough infections based on T cell responses. Likewise, there were no significant differences in the prediction of breakthrough infection severity based on NAb, T cell responses to Ag1, Ag3 or sum Ag1–3. The above findings were also confirmed using the multivariate models (Table 3). Furthermore, in the multivariate model, participants with T cell responses to Ag2 > 0.120 IU/mL had 31% higher odds of having mild-moderate limitations (*p* = 0.022).

In both the univariate and multivariate models, no statistically significant differences were observed between the odds of breakthrough infection or infection severity when the LOD of each assay was used as the threshold for the prediction model (Appendix A). No statistically significant differences were observed between the odds of breakthrough infection or infection severity when comparing different vaccines (Appendix A). The risk of breakthrough infections, based on the status of self-reported previous infections and/or positive anti-N results, was also analyzed. In univariate models, participants with negative anti-N results had 57% (*p* = 0.031) lower odds of having a breakthrough infection. However, self-reported previous infection alone or combined with anti-N were not significant predictors of breakthrough and the multivariate models did not confirm the significant finding about anti-N. 

## 4. Discussion

The AACC 2021 Annual Scientific Meeting & Clinical Lab Expo in September provided a useful opportunity to examine post-vaccine SARS-CoV-2 immune responses in a large cohort of volunteers that were diverse in areas such as their age, gender, race, ethnicity, vaccine types administered, and geography [11]. In this prospective study, we demonstrated that T cell responses to SARS-CoV-2 were uniformly durable up to 10 months during the study period, whereas the durability of NAb responses varied according to vaccine type. NAb responses to SARS-CoV-2 vaccination correlated with protection, and T cell memory responses may contribute to the protection against severe disease, but not against infection. 

Most of the previous studies and vaccine design/development strategies focused on maximizing the SARS-CoV-2 vaccine-induced humoral responses, particularly NAb [19]. The evidence for a protective role of NAb in SARS-CoV-2 infection includes preclinical animal models, passive immunization, first with convalescent patient sera and then with therapeutic monoclonal antibody preparations [20,21], and, finally, population studies on patients who have recovered from infection, or vaccination studies [19,22]. The lack of standardization of NAb assays, however, makes the comparison of these studies difficult and there is no consensus on the protective neutralization level against COVID-19 [19]. In addition, the decline in NAb levels raises concerns about the long-term effectiveness of SARS-CoV-2 vaccination. 

T cell-mediated immunity may be just as important in the long-lasting immune response to SARS-CoV-2 infection and vaccination as humoral responses. Virus-specific T cells were shown to be correlates of protective immunity against respiratory viruses such as the cause of the 2009 pandemic, H1N1 influenza [23]. T cell-mediated immunity also plays a role in the immune response to SARS-CoV-2 infection in preventing severe disease after natural infection [7]. In patients with humoral immune deficiency due to either genetic disease or undergoing B cell-depleting therapies, the value of T cell-mediated immunity is also clearly demonstrated [24,25]. T cell responses in resolving primary SARS-CoV-2 infection are also shown in experimental animal (mouse and macaque) infection models [20]. However, the contribution of T cells is difficult to analyze for both practical and conceptual reasons. First, there is no single way to measure T cell responses [7]. Second, on the practical side, most T cell assays require live cells, which often necessitates the isolation and storage of cells. Peripheral blood mononuclear cell isolation and cryopreservation introduces additional complexity and variability to the results. Because of these challenges, most large studies have omitted T cell testing, or it has been undertaken on only a small subset of samples. We utilized a recently available whole-blood interferon (IFN)-γ release assay for SARS-CoV-2, and evaluated a large cohort of participants. 

Our results confirmed previous findings [8,23,26,27,28] that the level of NAb gradually declines after vaccination (especially after Pfizer), whereas T cell responses were sustained up to 10–12 months post-vaccination for the three vaccine types. We also confirmed that NAb responses correlated with protection against infection but not disease severity. We also demonstrated that the T cell immunity responses did not predict breakthrough infection, but T cell responses to Ag2 indicated significant protection against self-reported severe COVID-19 disease. Protection against severe disease can only be detected using stimulants in QFN SARS-CoV-2 Ag2, but not Ag1 and 3. The Ag1 tube contains CD4+ T cell epitopes derived from the S1 subunit of the Spike protein, whereas the Ag2 tube contains both CD4+ and CD8+ T cell epitopes from the S1 and S2 subunits of the Spike protein. The Ag3 tube consists of CD4+ and CD8+ T cell epitopes from S1 and S2, plus immunodominant CD8+ T cell epitopes derived from whole genome. Although the exact composition of antigenic peptides is not available from the manufacturer, the main difference between Ag1 and 2 is the inclusion of CD8 epitopes, suggesting that detecting CD8 responses is the critical distinction. Developing a NAb response requires coordinated T and B cell responses. Helper CD4+ T cell responses are required for most NAb responses, therefore, NAb responses may partially indicate proper CD4 responses (“CD4 T helper” responses), but the cytotoxic CD8+ responses must be directly tested. Our results suggest that measuring these CD8 responses provides additional predictive information regarding disease severity. This finding is supported by experimental studies in macaques [29].

Translating these population-level risks to individual patients will remain challenging, however, as the outcome of SARS-CoV-2 infection depends on numerous individual host and pathogen variables. Viral variants are clearly critical, as was demonstrated by the emergence of Delta, Omicron, and other forms of SARS-CoV-2 that are different from the original strain. Although the T cell assay in the present study utilized the specific Ags derived from the full genome of the ancestral SARS-CoV-2 virus, but not of the VoCs, unlike serological response (and assays), the T cell responses are inherently resistant to viral mutations [3]. Accordingly, while mutations that escape antibody binding have commonly been documented for respiratory viruses such as influenza, and now for SARS-CoV-2, complete immune escape at the level of T cell responses has not been reported, to our knowledge, for any human acute respiratory infection. An additional feature of T cell biology is that whereas circulating preformed antibodies can directly and immediately bind to the virus for neutralization, T cell activation requires the processing of endogenously synthesized viral proteins, presented by the HLA molecules. This suggests that T cells may not prevent infection but may play a pivotal role in reducing viral load and lowering pathogenicity by eliminating the infected cells. Previous studies have shown that the majority of T cell responses are preserved against VoCs, suggesting that memory T cells may play a critical role against infection in light of the considerable escape of Omicron from NAb. It is congruent with our findings that T cell responses were associated with, and could predict the risk of, severe breakthrough infection, but not the prevention of the breakthrough infection. Studying whether SARS-CoV-2-specific T cell responses are more relevant to protection against or the prevention of severe diseases will provide important insights for creating better correlative studies investigating vaccine effectiveness, as well as will help define future vaccine strategies [7].

Our study has several limitations. First, the study only involved a single time point of sample collection, and the time of blood collection was not coordinated with the time of vaccination. However, the large sample size gave enough statistical power to perform longitudinal and multivariate analyses. Second, studying disease severity in contrast to infection rate requires a larger sample size [7] due to the low proportion of severe diseases. None of the study participants who reported a breakthrough infection required hospitalization. Third, the SARS-CoV-2 variant information of the breakthrough infection was not determined. However, since the breakthrough infections in our participants occurred during the time period of 9–23 February 2022, we speculate that the majority of those infections were caused by the Delta and Omicron variants [30]. Fourth, the assays used in this study targeted the original SARS-CoV-2 but not VoCs. Targeted assays may produce more specific results, garnering better associations with the clinical outcomes. In this study, the assays used tested stimulation-based responses rather than in vivo responses. Fifth, the definitions of breakthrough infection and limitations were self-reported in the follow-up survey, which could be subjective. Sixth, NAb responses were measured using a surrogate neutralization assay, and only measured the NAb binding to RBD; therefore, the breadth of responses was not measured. Finally, the anti-N serological responses may decline as well after natural infection, potentially decreasing the overlap between self-reported results and serological testing. 

Despite its limitations, our study proposes laboratory correlates for breakthrough infection and self-reported disease severity after SARS-CoV-2 vaccination. We showed that NAb responses can serve as predictors of breakthrough infection, whereas the T cell assay may aid in the prediction of breakthrough disease severity after SARS-CoV-2 vaccination.

## Figures and Tables

**Figure 1 viruses-15-00709-f001:**
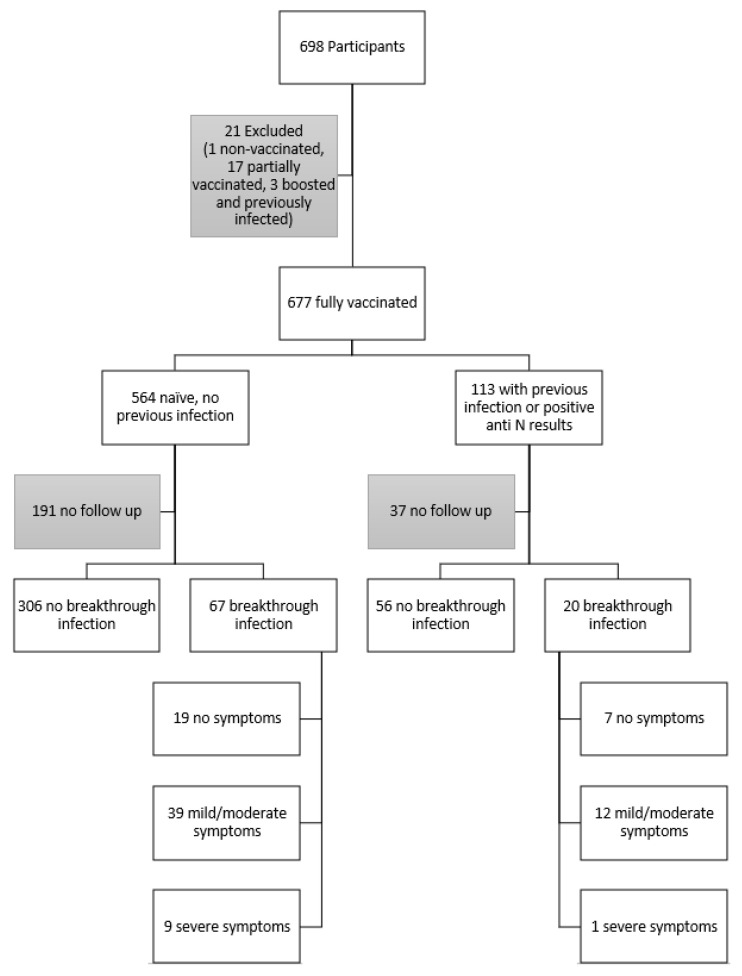
Consort chart for the subjects who had blood collected in September 2021. Breakthrough infections are defined as a follow-up positive COVID test reported in February 2022. Breakthrough infections and their severities are separated into those who had a previous COVID infection, and those who were infection-naïve at the time of blood draw.

**Figure 2 viruses-15-00709-f002:**
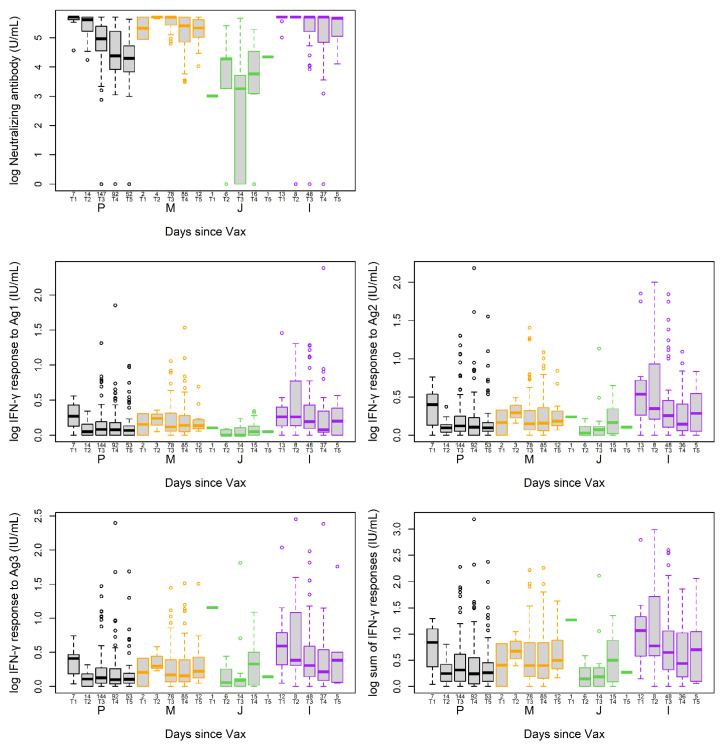
Distributions for T cell and NAb responses in participants receiving different SARS-CoV-2 vaccines. Each box shows the IQR with the median as the interior bar; the outer bars stretch out to the minimum and maximum, with the exception that points outside the fence are shown as individual circles. P: Pfizer; M: Moderna; J: Johnson & Johnson; I: previous infection or positive anti-N; T1: 0–2 months; T2: 2–4 months; T3: 4–6 months; T4: 6–8 months; T5: 8–10.5 months. Numbers indicated the time period indicators (e.g., T1–T5) are the number of participants for that group.

**Table 1 viruses-15-00709-t001:** Participant characterization and demographics.

Demographic ^#^		Blood Draw (*n* = 698)	No Known Previous Infection and Negative Anti-N				Self-Reported Previous Infection or Positive Anti-N (*n* = 113)		*p* Value
			Pfizer (*n* = 314)	Moderna (*n* = 181)	Johnson & Johnson (*n* = 38)	Booster (*n* = 31)		Positive Anti-N * (*n* = 46)	
Sex	Male, N (%)	328 (46.9)	141 (44.9)	89 (49.2)	17 (44.7)	12 (38.7)	60 (53.1)	28 (60.9)	0.48
Age	Median (IQR)	50 (40–59)	50 (39–58)	50 (40–60)	49 (38–58)	50 (40–66)	49 (40–58)	49 (40–58)	0.66
Race	Caucasian, N (%)	528 (75.6)	239 (76.1)	137 (75.7)	30 (78.9)	25 (80.6)	89 (78.8)	37 (80.4)	0.86
	African American /Black, N (%)	41 (5.8)	20 (6.4)	10 (5.5)	3 (7.9)	0 (0.0)	6 (5.3)	4 (8.7)	
	Asian, N (%)	67 (9.5)	30 (9.6)	18 (9.9)	3 (7.9)	2 (6.5)	9 (8.0)	4 (8.7)	
	Native Hawaiian /Pacific Islander, N (%)	6 (0.8)	3 (1.0)	3 (1.7)	0 (0.0)	0 (0.0)	0 (0.0)	0 (0.0)	
	Unknown/Other /Prefer Not to Say, N (%)	56 (8.0)	22 (7.0)	13 (7.2)	2 (5.2)	4 (12.9)	9 (8.0)	1 (2.2)	
Ethnicity	Hispanic and/or Latino, N (%)	92 (13.1)	33 (10.5)	22 (12.2)	6 (15.8)	5 (16.1)	22 (19.5)	5 (10.9)	0.56
	Non-Hispanic /Non-Latino, N (%)	575 (82.3)	268 (85.4)	151 (83.4)	30 (78.9)	25 (80.6)	87 (77.0)	38 (82.6)	
	Prefer not to reply, N (%)	31 (4.4)	13 (4.1)	8 (4.4)	2 (5.3)	1 (3.2)	4 (3.5)	3 (6.5)	

* specimens from 653 participants were measured for anti-N antibodies. ^#^ All blood draws are presented in the first column and 677 non-excluded subjects are separated in the other columns. The *p*-value compares five groups: the three vaccine-only groups, the boosted group, and the vaccinated and previously infected group.

**Table 2 viruses-15-00709-t002:** Median (IQR) values for the measured assays for participants with or without breakthrough infections including all vaccination and previous infection statuses.

449 of the 677 Completed the Follow-Up Questionnaire	No Breakthrough Infection (N = 362)	Breakthrough Infection (N = 87)	Among Those with Breakthrough			Uni * *p* Value	Multi *p* Value
			No Limitation (*n* = 26)	Mild-Moderate Limitations (*n* = 51)	Severe Limitations (*n* = 10)		
NAb (%)	88 (69–95)	83 (57–96)	87.5 (70.5–97)	82 (55.5–93.5)	72 (53–85.8)	0.475	0.405
NAb (UmL)	194.9 (86.3–300)	151.4 (58.7–300)	185.7 (74.8–300)	148.4 (50.4–285.8)	91.6 (46.9–198.9)	0.285	0.197
Ag1, median (IQR)	0.11 (0.04–0.30)	0.11 (0.03–0.30)	0.09 (0.02–0.28)	0.14 (0.04–0.30)	0.11 (0.01–0.41)	0.547	0.572
Ag2, median (IQR)	0.16 (0.06–0.40)	0.16 (0.05–0.45)	0.15 (0.05–0.36)	0.20 (0.05–0.47)	0.08 (0.01–0.19)	0.629	0.621
Ag3, median (IQR)	0.17 (0.06–0.50)	0.18 (0.06–0.48)	0.17 (0.05–0.51)	0.19 (0.07–0.48)	0.17 (0.01–0.29)	0.873	0.874
Sum, median (IQR)	0.46 (0.16–1.24)	0.51 (0.12–1.10)	0.47 (0.11–1.05)	0.54 (0.15–1.11)	0.37 (0.04–0.99)	0.766	0.762

NAb = Neutralizing Antibody, Ag1–3 = T cell response to Ags 1–3 (IU/mL), Sag = sum of T cell responses to Ags1–3 (IU/mL). * The univariate *p*-value compares assay levels among different degrees of limitation during infection. The multivariate *p*-value compares the same three groups, while adjusting for age, sex, time from vaccine to blood draw, previous infection status before blood draw, time after blood draw to breakthrough infection.

**Table 3 viruses-15-00709-t003:** Risk of breakthrough infections using thresholds chosen based on ROC analysis.

Univariate						
COVID-19	*n*	NAb (>95 U/mL)	sum Ag(>0.385 IU/mL)	Ag1 (>0.435 IU/mL)	Ag2 (>0.120 IU/mL)	Ag3 (>0.055 IU/mL)
No breakthrough	362	**1.78**	0.88	1.15	0.89	1.22
No limitations	26	1.07	0.95	0.84	0.93	0.90
Mild-moderate limitations	51	1.03	1.15	1.04	1.23	1.22
Severe limitations	10	0.91	0.92	1.14	**0.87**	0.91
Hospitalizations	0	-	**-**	**-**	**-**	**-**
Multivariate						
COVID-19	*n*	NAb (>95 U/mL)	sum Ag(>0.385 IU/mL)	Ag1 (>0.435 IU/mL)	Ag2 (>0.120 IU/mL)	Ag3 (>0.055 IU/mL)
No breakthrough	362	**1.95**	0.91	1.21	0.92	1.27
No limitations	26	1.04	0.91	0.85	0.90	0.88
Mild-moderate limitations	51	1.08	1.23	1.03	**1.31**	1.28
Severe limitations	10	0.90	0.89	1.14	**0.85**	0.89
Hospitalizations	0	-	**-**	**-**	**-**	**-**

Multivariate models also adjust for age, sex, time since last vaccine of blood draw, previous infection status at blood draw, and time between blood draw and infection. Values are (adjusted) odds ratios for the event in the second column. Significant changes from the reference time are in bold with *p* < 0.05.

## Data Availability

Data is available upon request.

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
