# Peer review of "T Cell Responses Correlate with Self-Reported Disease Severity and Neutralizing Antibody Responses Predict Protection against SARS-CoV-2 Breakthrough Infection"

_viruses, 2023, doi:10.3390/v15030709_

Round 1
Reviewer 1 Report
The rapid speed at which COVID-19 vaccines were produced and distributed was unprecedented. While these vaccines have done an amazing job at reducing death and mortality from SARS-CoV-2 infection, there is still a lot to learn about the vaccine-induced immune response generated. This information will be critical for determining the most appropriate vaccination strategies moving forward. Thus, it is important to continue to conduct studies like this to collect data on immune responses to COVID-19 vaccines.
In this study, Zhao et al conduct a prospective study with 677 participants to compare neutralising antibodies and T cell responses to three different COVID-19 vaccines. In addition, a follow up study was performed 4 months after the initial blood draw to report on breakthrough infections. The authors identified a correlation with Nab titres and risk of breakthrough infection. Of the participants who became infected, those that experienced severe symptoms, had lower T cell responses (against Ag 2 which included both CD4 and CD8 epitopes). There are some limitations to this study, for example, time of blood collection was not coordinated with time of vaccination, Nab titers were measured using a surrogate neutralisation assay and only measured the NAbs binding to RBD and therefore, the breadth of response was not measured. The specificity of the immune response was targeted towards the ancestral strain (one assumes this, the details of the QuantiFERON antigens were not disclosed), however, measuring responses against VOCs may be more relevant to the strains involved in the breakthrough infection (omicron, delta). One of the key findings was based on a very small sample size (N=10 breakthrough, severe) specific to one type of Ag (Ag2 – where specific details on the epitopes are not disclosed). Nonetheless, the information reported in this study is relevant and informative.
Minor points
1) The method section on study design and statistical modelling could include more information. The description on the criteria for participants could have be clearer and all relevant information on study design could have been included in one paragraph/subheading. Figure 1 was helpful to follow study design, however in reference to the individuals excluded: 3 previously infected, it wasn’t clear the reasoning for exclusion – did these participants also receive a booster?
2) There is little information/explanation on the statistical models used to correct for time after last vaccine dose and correlation with breakthrough infection. It would be helpful to understand how the corrections were performed. Although, it was nice to see the values from the individual times after last vaccine doses in figure 2.
3) Is it possible to provide more information on the booster group, which vaccine was given, were they a mix of heterologous or homologous boosts?
4) Were the 19 boosted individuals that had a breakthrough infection separated out from the analysis in table 2? Please clarify.
5) Please double check the information provided in the discussion section referring to the epitopes tested, “All of the vaccines in this study are based on S1, therefore the inclusion of non-S1 peptides is not relevant for measuring the response to these vaccines (although they can be useful after natural infection).”
These vaccines encode the full-length spike – and therefore will elicit both S1 and S2 specific responses.
Author Response
Reviewer 1
The rapid speed at which COVID-19 vaccines were produced and distributed was unprecedented. While these vaccines have done an amazing job at reducing death and mortality from SARS-CoV-2 infection, there is still a lot to learn about the vaccine-induced immune response generated. This information will be critical for determining the most appropriate vaccination strategies moving forward. Thus, it is important to continue to conduct studies like this to collect data on immune responses to COVID-19 vaccines.
In this study, Zhao et al conduct a prospective study with 677 participants to compare neutralising antibodies and T cell responses to three different COVID-19 vaccines. In addition, a follow up study was performed 4 months after the initial blood draw to report on breakthrough infections. The authors identified a correlation with Nab titres and risk of breakthrough infection. Of the participants who became infected, those that experienced severe symptoms, had lower T cell responses (against Ag 2 which included both CD4 and CD8 epitopes). There are some limitations to this study, for example, time of blood collection was not coordinated with time of vaccination, Nab titers were measured using a surrogate neutralisation assay and only measured the NAbs binding to RBD and therefore, the breadth of response was not measured. The specificity of the immune response was targeted towards the ancestral strain (one assumes this, the details of the QuantiFERON antigens were not disclosed), however, measuring responses against VOCs may be more relevant to the strains involved in the breakthrough infection (omicron, delta). One of the key findings was based on a very small sample size (N=10 breakthrough, severe) specific to one type of Ag (Ag2 – where specific details on the epitopes are not disclosed). Nonetheless, the information reported in this study is relevant and informative.
Response: Thank you for carefully reading our manuscript and providing very helpful suggestions. We acknowledge and agree with the limitations that were pointed out by the reviewers. In addition to the limitations described in the original submission, we have modified the discussion to include the limitations suggested by the reviewer at line 435 and line 450. Unfortunately, certain technical details regarding the QuantiFERON antigens and specific details on the Ag2 epitopes were not available since the information was not disclosed by the manufacturers.
Minor points
1) The method section on study design and statistical modelling could include more information. The description on the criteria for participants could have be clearer and all relevant information on study design could have been included in one paragraph/subheading. Figure 1 was helpful to follow study design, however in reference to the individuals excluded: 3 previously infected, it wasn’t clear the reasoning for exclusion – did these participants also receive a booster?
Response: We have consolidated the exclusion and inclusion criteria to one paragraph at line 113. Yes, the 3 excluded people were previously infected and had been vaccinated and received a booster shot. The description of those 3 individuals in figure 1 has been expanded for clarity.
2) There is little information/explanation on the statistical models used to correct for time after last vaccine dose and correlation with breakthrough infection. It would be helpful to understand how the corrections were performed. Although, it was nice to see the values from the individual times after last vaccine doses in figure 2.
Response: The covariates included in the multivariate models have been added to the model descriptions at lines 201 and 213. Multivariate models include age, sex, time from vaccine to blood draw, previous infection status before blood draw, and time after blood draw to breakthrough infection.
3) Is it possible to provide more information on the booster group, which vaccine was given, were they a mix of heterologous or homologous boosts?
Response: More information was added at line 227.
4) Were the 19 boosted individuals that had a breakthrough infection separated out from the analysis in table 2? Please clarify.
Response: Table 2 includes individuals with all vaccination types, including those who were boosted. We have clarified that in the caption for Table 2.
5) Please double check the information provided in the discussion section referring to the epitopes tested, “All of the vaccines in this study are based on S1, therefore the inclusion of non-S1 peptides is not relevant for measuring the response to these vaccines (although they can be useful after natural infection).”
These vaccines encode the full-length spike – and therefore will elicit both S1 and S2 specific responses.
Response: Thank you for pointing it out. The cited sentence has been deleted.

Reviewer 2 Report
The article by Zhen Zhao et al. presents the role of adaptive immunity in response to SARS-CoV-2 various vaccines in large, diverse populations. The authors found that NAb levels decreased with time after vaccination with Pfizer and Johnson & Johnson. T cell responses showed no significant difference among the different vaccines and remained stable up to 10 months of the study period for all vaccine types. NAb responses (< 95 U/mL) predicted breakthrough infection, whereas previous infection, type of vaccine, and T cell responses did not. T cell responses to viral epitopes (< 0.120 IU/mL) showed a significant association with the self-reported severity of COVID-19 disease. This is an important study that has clinical relevance.
Author Response
Reviewer 2
The article by Zhen Zhao et al. presents the role of adaptive immunity in response to SARS-CoV-2 various vaccines in large, diverse populations. The authors found that NAb levels decreased with time after vaccination with Pfizer and Johnson & Johnson. T cell responses showed no significant difference among the different vaccines and remained stable up to 10 months of the study period for all vaccine types. NAb responses (< 95 U/mL) predicted breakthrough infection, whereas previous infection, type of vaccine, and T cell responses did not. T cell responses to viral epitopes (< 0.120 IU/mL) showed a significant association with the self-reported severity of COVID-19 disease. This is an important study that has clinical relevance.
Response: Thank you for carefully reading our manuscript and providing very positive comments on the importance and clinical relevance of our study.